# Lichen Planus Activity and Damage Index (LiPADI)–Creation of the Questionnaire

**DOI:** 10.3390/jcm11010023

**Published:** 2021-12-22

**Authors:** Katarzyna Stępień, Ewa Żabska, Mansur Rahnama-Hezavah, Adam Reich

**Affiliations:** 1Department of Dermatology, Institute of Medical Sciences, Medical College of Rzeszow University, 35-055 Rzeszow, Poland; kat_st@o2.pl (K.S.); e.zabska@wp.pl (E.Ż.); 2Chair and Department of Dental Surgery, Medical University of Lublin, 20-093 Lublin, Poland; mansur.rahnama@umlub.pl

**Keywords:** assessment tool, disease severity, lichen planus

## Abstract

Introduction: Lichen planus (LP) is a chronic autoimmune disease that affects skin, oral and genital mucosa, and other sites. Basic difficulties in assessment of LP are multitude of disease forms and diverse locations of lesions. Moreover, there is lack of objective and consolidated tool for assessment of disease severity and LP progression. Objective: The aim of the study was to develop a valid evaluation tool of LP severity, which will enable disease assessment in a repetitive way. Materials and methods: A combined tool called Lichen Planus Activity and Damage Index (LiPADI) was developed to assess the severity of LP skin, mucosal, and nail lesions as well as hair loss/scaring alopecia to provide an integrative scoring for LP activity and damage caused by the disease. Skin lesions were assessed in nine locations: scalp, face, chest, abdomen, back and buttocks, arms, hands, legs, and feet. The assessment of lesion activity included erythema, hypertrophy, and scaling, while the damage was reflected by the assessment of hyperpigmentation and scaring/atrophy. In addition, mucosal lesions, nail abnormalities, hair loss, and scarring alopecia were evaluated as well. LiPADI scoring was compared with quality of life assessed with the Dermatology Life Quality Index, EQ-5D calculator, pain and pruritus intensity assessed with the Numerical Rating Scale as well as with the patient and physician global assessment. Results: Our results show that LiPADI well reflects the LP patient’s clinical condition. The obtained results were in line with other indicators assessed. In addition, it was possible to evaluate patients with various forms and locations of LP, what indicates its versatility. Conclusions: LiPADI seems to be a useful tool for measurement the severity of the LP and its progress over time, which could help to monitor the effectiveness of the patients’ treatment.

## 1. Introduction

Lichen planus (LP) is a chronic inflammatory mucocutaneous disease that affects about 0.5–2% of the general population [1]. The clinical presentation varies depending on the variant of LP and area involved, which may be skin, scalp, nails, and mucous membranes. Typical cutaneous LP is characterized by flat-topped, polygonal, violaceous papules and plaques [2]. Usually, the lesions are located on the trunk and flexor surfaces of the forearms, wrists, and ankles. While resolving, the lesions may also result in long-standing, residual hyperpigmentation. Some patients may demonstrate other clinical subtypes of LP, like hypertrophic, bullous, pigmented, linear, annular, atrophic, hypertrophic, inverse, eruptive, ulcerative, or actinic LP, LP pemphigoides, and overlap syndromes [2]. Clinical presentation of the rarer subtypes of LP may be dissimilar to classic LP; however, histopathological examination reveals consistent features, including band-like subepidermal lymphocytic inflammatory infiltrate and degeneration of the basal cell layer of the epidermis [2,3]. Lichen planopilaris is considered as a follicular form of LP and classically presents as perifollicular erythema and scaling. This form of LP leads to irreversible destruction of hair follicle stem cells and to scarring alopecia [4]. When the nails are affected by LP, longitudinal ridging is the most common finding. Progression of the nail disease may result in dorsal pterygium and permanent nail destruction [5].

The most common affected mucosal sites for LP are oral and genital mucosae. Usually, mucosal lesions demonstrate more chronic course than typical skin lesions. Oral LP is characterized by symmetric reticular whitish lines as well as by grouped papules and erosions on the buccal mucosa, tongue, soft palate, gingiva, and lips. Vulvovaginal LP most commonly affects postmenopausal women, and may present as erosions surrounded by white lacy border and whitish areas [6]. On the glans penis LP may present clinically as erythematous macules, papules, lacy network, and atrophic lesions [7].

The objective assessment of any disease severity is of great importance in dermatology and other medical specialties. Severity scales are essential to evaluate treatment outcomes in clinical trials and to compare results of various studies. Furthermore, such scales are invaluable in clinical practice for overall judgement of disease severity, selecting a therapeutic option and then guiding the course of treatment. Therefore, the measurements play important role in the therapeutic and research process. There are numerous severity scales available to evaluate various dermatologic diseases, the most widely used of which are e.g., SCORing Atopic Dermatitis (SCORAD) and Eczema Area and Severity Index (EASI) for assessment of atopic dermatitis [8], Psoriasis Area and Severity Index (PASI) and Body Suface Area (BSA) for assessment of psoriasis [9], and The Localized Scleroderma Assessment Tool (LoSCAT) for assessment of the severity of localized scleroderma (morphea) [10]. However, due to the variety of clinical manifestations and the coexistence of lesions in different locations, comprehensive assessment of the severity of LP remains a challenge. To date, no appropriate tool for the overall assessment of LP has been proposed. Therefore, the objective of this study was to develop and validate scoring system to assess severity of LP in terms of activity and damage, taking into consideration various clinical features and different variants of the disease.

## 2. Methods

### 2.1. Subjects

A total of 44 patients (36 females and 8 males) aged between 16 and 80 years (mean age: 52.8 ± 15.6 years) with different types of biopsy-confirmed LP were included in the study. The patients were recruited from inpatients and outpatients of the Department of Dermatology of the University of Rzeszow, Poland. All patients were Caucasians. Seven (15.9%) lived in a big city (>100,000 inhabitants), 14 (31.8%) in a town, and the rest (*n* = 23, 52.3%) in small villages. Regarding education, 9 (20.4%) patients had primary school, 5 (11.4%) had secondary school, 15 (34.1%) finished high school, and the remaining 15 (34.1%) finished university. All patients agreed to participate in the study and signed the written informed consent.

### 2.2. Study Design

After providing a written informed consent, all participants were asked about their demographics and the course of the disease. The type of LP was classified based on the appearance of the lesions. Disease severity was measured independently using Lichen Planus Activity and Damage Index (LiPADI) by two independent physicians. Next, all included subjects were asked to complete Dermatology Life Quality Index (DLQI) [11], 12-Item Pruritus Severity Scale (12-PSS) [12], EQ-Visual Analogue Scale (EQ-VAS), and to record their pruritus and pain intensity using the Numerical Rating Scale (NRS) [13]. In addition, physician’s (PGA) and patient’s subjective global assessment (PtGA) of disease severity (mild, moderate, or severe) was obtained.

### 2.3. Development of Lichen Planus Area and Damage Index

LiPADI was modeled based on the Cutaneous Lupus Erythematosus Disease Area and Severity Index (CLASI) [14]. LiPADI as well as CLASI is designed as a table in which rows contain anatomical areas, while columns include scores for clinical symptoms (Appendix A). Disease activity and skin damage are scored separately in multiple anatomic locations, without measuring the proportion of involved skin. Using LiPADI, activity is scored in terms of erythema, hypertrophy, scale, mucous membrane involvement, acute hair loss, and non-scarring alopecia, whereas damage done by the disease is measured by assessing hyperpigmentation, scarring atrophy, nail lesion abnormalities, and scarring of the scalp. The scores for disease activity and damage are calculated separately by simple addition. The extent of involvement for each of the specific anatomic areas is scored according to the worst affected lesion.

### 2.4. Statistics

Statistic analyses were performed with Statistica software (Statsoft Polska, Kraków, Poland). Descriptive analyses were done to assess distribution. Spearman correlation was used to determine correlation coefficients between pairs of variables: (1) LiPADI variables (activity and damage) and (2) Physician- and patient-reported variables (DLQI, 12-PSS, EQ-VAS, pruritus and pain NRS, physician and patient global assessment). The normal distribution was verified with Kolgomorov–Smirnov test. A *p*-value < 0.05 was considered statistically significant.

## 3. Results

### 3.1. Lichen Planus Characteristics

Regarding the clinical subtypes of LP [15], following variants were diagnosed in recruited patients: 19 (43.2%) cases of lichen planopilaris, 16 (36.4%) eruptive guttate LP, 6 (13.6%) classic popular LP, 4 (9.9%) annular LP, 3 (6.8%) erosive LP, 1 (2.3%) hypertrophic LP, 1 (2.3%) atrophic LP, and 1 (2.3%) bullous LP. In addition, 2 (4.5%) patients had both guttate LP and palmoplantar LP, and 1 (2.3%) patient had palmoplantar LP coexisting with lichen planopilaris. The mucosal lesions were present in 21 (47.7%) patients and nail abnormalities typical for LP were found in 20 (45.5%) subjects. The mean disease duration was 4.2 ± 5.6 years (range: 1 month–29 years) and the mean duration of the current disease episode was 10.9 ± 14.9 months (range: 1 week–6 years).

### 3.2. Distribution of LiPADI

Theoretically, the scoring of LiPADI ranges from 0 to 71 points for activity and for 0 to 37 points for damage; however, it is hardly possible that one patient will get the maximum value in all categories at one time point. Among the 44 participants, the total scoring for activity was between 1 and 44 points (mean: 12.3 ± 9.6 points, median: 9 points) and for damage between 0 and 12 (mean: 4.2 ± 2.6 points, median: 4 points). There was some bottom effect observed for both activity and damage, as about one-third of patients received ≤5 points for activity (*n* = 15, 34.1%) and more than half of patients had damage scoring ≤4 points (*n* = 26, 59.1%). However, still the total scoring for activity and damage had normal distribution according to Kolgomorov–Smirnov test (d = 0.16, *p* > 0.2, and d = 0.15, *p* > 0.2, respectively).

### 3.3. Interrater Reliability and Internal Consistency of LiPADI

The consistency of a measure across rater was high. No significant differences were observed between two independent assessors of LP severity (activity scoring: 13.4 ± 9.6 points vs. 13.8 ± 10.5 points, *p* = 0.28; damage scoring 4.2 ± 2.7 points vs. 4.2 ± 3.1 points, *p* = 0.93). In addition, there were also no differences between the two measurements with regard to any domain of activity or damage scoring (Figure 1). The intraclass correlation coefficient (ICC) for activity scoring of LiPADI was 0.96, and ICC for damage scoring was 0.8.

Calculation of the Cronbach α coefficient revealed that the internal consistency of the LiPADI components of LP activity was strong (Cronbach α = 0.85) and that of LP damage was satisfactory (Cronbach α = 0.6).

### 3.4. Discriminant and Convergent Validity of LiPADI

While comparing LiPADI scoring with the global disease severity assessed by physicians (PGA), a significant correlation was observed between the LiPADI activity scoring and PGA (ρ = 0.51, *p* < 0.001), while no significant correlation was noted between LiPADI damage and PGA (ρ = 0.19, *p* = 0.22). Scoring of LiPADI activity significantly differed between patients with mild, moderate, and severe LP (*p* < 0.01) (Figure 2). Importantly, no significant correlation was observed between LiPADI scoring and PtGA (ρ = 0.1, *p* = 0.56 for activity, ρ = 0.19, *p* = 0.22 for damage). Nevertheless, significant differences between each LP severity level assessed by physicians and LiPADI activity score documented that LiPADI possesses the ability to differentiate various subgroups of LP patients.

Considering other measurements performed in LP patients, we observed significant correlations between LiPADI Activity scoring and DLQI scoring (ρ = 0.38, *p* = 0.01), as well as pruritus intensity assessed both by 12-PSS (ρ = 0.41, *p* = 0.01) and NRS (ρ = 0.5, *p* = 0.001), while LiPADI damage scoring correlated significantly with disease duration (ρ = 0.3, *p* < 0.05) (Table 1).

## 4. Discussion

Increasing interest in the development of new therapies for many dermatological conditions in recent years implies the need for more accurate and reliable measurement of treatment outcomes. Thus, various assessment instruments are being prepared and validated for different skin disease, e.g., for psoriasis or atopic dermatitis. However, in some entities the assessment might be challenging due to numerous clinical manifestations, different subjective sensations, and variable clinical course with some persistent damages in selected patients. To our knowledge, the development of LiPADI is the first attempt to comprehensively assess the severity of LP in various locations, with particular emphasis on the skin.

Numerous scoring systems for the assessment of LP subtypes have been reported in the past; however, none of them have been recognized to date as a gold standard. For instance, more than 22 specific scoring systems were designed to grade the severity of oral LP [16], but these scales do not consider cutaneous lesions and have limited applicability for most LP patients, as they often present skin lesions. Of note, neither of these scales is universally accepted even for oral lesions, although the Thongprasom scoring system is most commonly used in clinical trials [17]. Similar situation may be observed in lichen planopilaris in which there have been some instruments developed for the assessment of disease severity [18,19,20], but they can be used only regarding lesions located within the skull area.

Interestingly, only two scoring systems have been proposed for cutaneous LP so far. In 2019, Bishnoi et al. [21] proposed Lichen Planus Activity, Area and Severity Index (LPAASI). The LPAASI has two components, one component which defines disease activity in terms of progression of existing lesions and appearance of new lesions, and the second one which defines disease severity based on the morphology of the lesions, and extent of the disease considering the body surface area involved by LP [21]. Another scoring system for LP—Lichen Planus Severity Index (LPSI)—published in 2020, requires calculation of the involved skin area using the rule of nines and lesion counting. To determine lichen planus severity, the counting of total number of lesions and assessing the percentage of each of the morphological types (erythematous papules, violaceous papules, violaceous plaques, hyperpigmented hypertrophic papules and plaques, and post-inflammatory hyperpigmentation) is needed, followed by assignment of the area involvement factor. The final scoring requires multiplication with the multiplying factors for each respective lesion type and the total BSA factor [22]. It has to be mentioned that both scales are quite complicated, time consuming, and difficult for implementation in routine daily practice, and only LPSI has been validated to some extend so far [21,22].

Here, we have proposed a new assessment tool for LP (LiPADI) which has been developed based on the scoring system of cutaneous lupus erythematosus (CLE)—CLASI—as, in our opinion, clinical manifestations of LP have a lot to do with the clinical manifestations of CLE. We have also provided some provisional data on the validity of LiPADI, but due to a limited number of analyzed patients and underrepresentation of some subtypes of LP, these results have to be considered with caution and need to be verified in the future studies. We would like to underline the necessity to verify the utility of the scale particularly in LP limited to the mucous membranes, including both erosive and reticular/papular variants.

Nevertheless, we hope that it will be found easier and more comprehensive than previously developed instruments and may be implemented both in clinical trials as well as in routine daily practice. Similar to the assessment of CLE with CLASI, we divided the LP symptoms into two groups, namely, activity and damage, as we believe that some lesions in LP are just the consequences of the inflammatory process but do not necessarily reflect the current disease severity. However, they cannot be omitted, as it is important to have the comprehensive impact of the disease assessed in all patients. The essence of such division could be supported by the observations that damage scoring did not correlate with any assessments used in our study except for disease duration which can be easily explained by the fact that the longer the disease lasts, the more damage it causes. In contrast, the activity scoring significantly correlated with many aspects analyzed in our patients, such as global disease severity, quality of life, or pruritus severity. No significant difference of LiPADI activity scoring between various levels of lichen planus severity assessed by patients may be explained by the fact that such assessment done by the patient is very subjective. Sometimes, even limited lesions, e.g., lichen planopilaris, may be considered as severe, because they may produce significant suffering due to pain or pruritus, and uncertainty caused by frequently unsuccessful therapies. We have also demonstrated that LiPADI is characterized by a good internal consistency and interrater reliability, which further supports its clinical applicability.

It has to be underlined that our study contains also several limitations. The major limitation is rather small number of analyzed patients and we hope to continue the validation in the future, as assessing the validity of any scale is always an ongoing and long-lasting process. Furthermore, the studied population was not balanced regarding particular LP subtypes, and some variants were overrepresented, while many others were even absent. However, despite LP demonstrating significant clinical variability, many clinical variants are rare and it would be impossible to collect a representative group of patients with all clinical variants over a reasonable period of time. Finally, it would be valuable to further test the intra-rater reliability among larger group of physicians to know the true variability of achieved results. Despite these limitations, we still consider that the new instrument is of value and worth being further tested in next clinical trials.

In conclusion, the LiPADI has been proved to be an effective system for assessment disease severity in lichen planus showing satisfactory convergent validity, good test–retest reproducibility, high internal consistency, and some discriminating properties. We hope that, in the near future, LiPADI will be used also by other researchers.

## Figures and Tables

**Figure 1 jcm-11-00023-f001:**
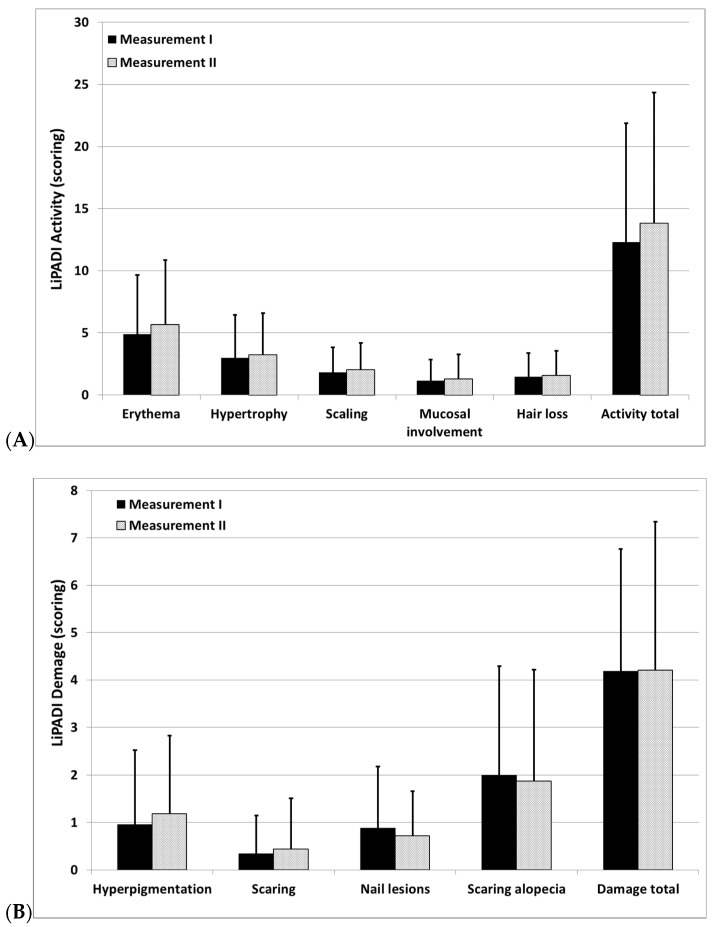
(**A**) Comparison of the scoring of total LiPADI activity and its subdomains performed by two independent assessors. (**B**) Comparison of the scoring of total LiPADI damage and its subdomains performed by two independent assessors (results demonstrates as means ± standard deviations, *p* > 0.05 for all comparisons).

**Figure 2 jcm-11-00023-f002:**
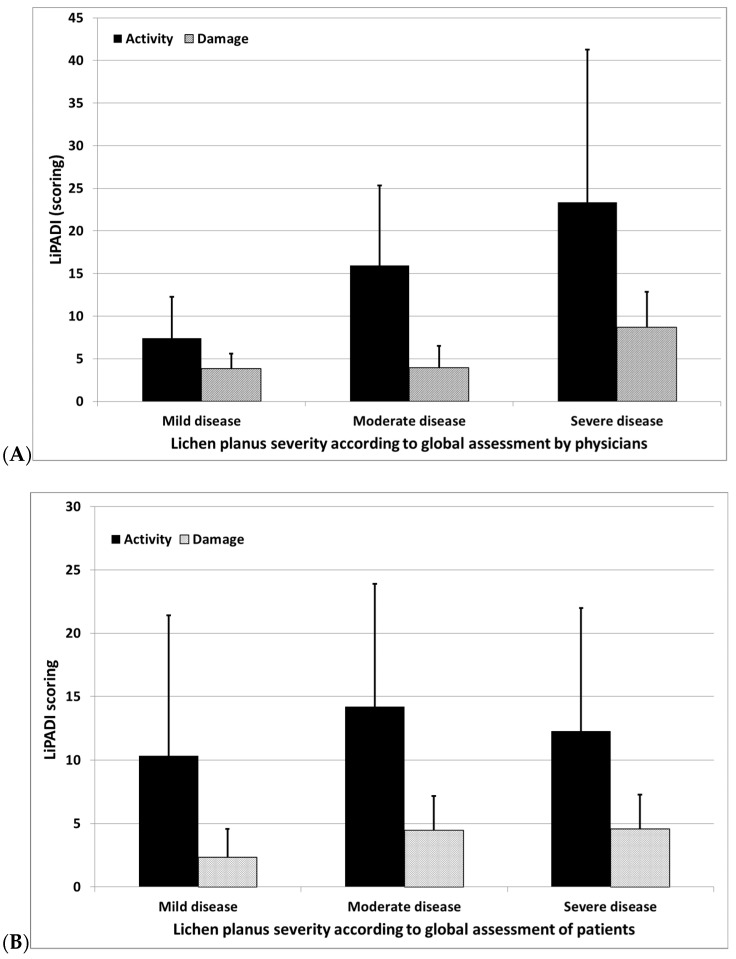
(**A**) Comparison of LiPADI activity and damage scoring between various levels of lichen planus severity assessed by physicians (*p* < 0.01 for activity, *p* = 0.16 for damage). (**B**) Comparison of LiPADI activity and damage scoring between various levels of lichen planus severity assessed by patients (*p* = 0.49 for activity, *p* = 0.22 for damage) (results demonstrates as means ± standard deviations).

**Table 1 jcm-11-00023-t001:** Correlations coefficients between the domains of LiPADI (Lichen Planus Activity and Damage Index) with other tested variables (*p* values according to Spearman rank correlation test) (DLQI—Dermatology Life Quality Index, EQ-VAS—EQ-Visual Analogue Scale EQ-5D, NRS—Numerical Rating Scale, 12-PSS—12-Item Pruritus Severity Scale).

	LiPADI	
	Activity	Damage
DLQI	ρ = 0.38, *p* = 0.01	ρ = −0.08, *p* = 0.63
12-PSS	ρ = 0.41, *p* = 0.01	ρ = −0.17, *p* = 0.32
EQ-VAS	ρ = −0.15, *p* = 0.34	ρ = −0.08, *p* = 0.63
NRS pruritus	ρ = 0.5, *p* = 0.001	ρ = −0.08, *p* = 0.6
NRS pain	ρ = 0.09, *p* = 0.6	ρ = 0.1, *p* = 0.53
Disease duration	ρ = −0.28, *p* = 0.06	ρ = 0.3, *p* < 0.05
Duration of exacerbation	ρ = −0.3, *p* < 0.05	ρ = −0.02, *p* = 0.9

## Data Availability

The data presented in this study are available on request from the corresponding author. The data are not publicly available due to personal data protection rules at our Institution.

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
