# Peer review of "Lichen Planus Activity and Damage Index (LiPADI)–Creation of the Questionnaire"

_jcm, 2021, doi:10.3390/jcm11010023_

Round 1

Reviewer 1 Report

The idea of developing a comprehensive scale to measure LP severity is not a new one. The authors introduced a new scale with different components. The idea is great but it might not be used in real life. If the publisher allows, I suggest that all the forms and tools used in the LiPADI scale be added to the article and be made downloadable. I know the authors pointed to the tools in the reference section, remember that not all practitioners have access to download the references. 

Author Response

We are grateful to the reviewer for his/her very supportive comments. We have added the questionnaire as an appendix to the manuscript. As the manuscript will be open access, all readers can download and use this questionnaire. 

Reviewer 2 Report

LP is not an infrequent disease and although some forms are really rare, such as the bullous and atrophic forms; the erosive form is relatively frequent. I do not understand why, however, the authors only included one patient from this cohort. Also, the fact that many of the forms only include one patient makes me really think about the validity of this scale. Maybe the scale should be used for hypertrophic forms. Please add a paragraph about this issue.

I believe that the number of patients should be increased to be able to validate this scale. For this, it is necessary to modify IN THE TITLE saying that it is a proposal that must be validated.

Author Response

We are grateful to the reviewer for his/her valuable comments. Please find below our answers to the queries: 

Reviewer:

LP is not an infrequent disease and although some forms are really rare, such as the bullous and atrophic forms; the erosive form is relatively frequent. I do not understand why, however, the authors only included one patient from this cohort. Also, the fact that many of the forms only include one patient makes me really think about the validity of this scale. Maybe the scale should be used for hypertrophic forms. Please add a paragraph about this issue.

Authors' response: We agree with the reviewer, that some variants of lichen planus are underrepresented in our cohort of patients. A low number of erosive LP could be explained by the fact, that such patients are rarely referred and hospitalized in our clinic. In Poland, patients with erosive lichen planus limited to the mouth are usually treated in the departments of periodontologists.  Furthermore, a number of scales exist to precisely assess the severity of lesions limited to the oral mucosa. Our major goal was to establish a new severity scale, that could comprehensively assess both skin, hair, and mucosa lesions. 

In the revised manuscript we have added a short paragraph discussing this issue (marked in red). 

Reviewer:

2. I believe that the number of patients should be increased to be able to validate this scale. For this, it is necessary to modify IN THE TITLE saying that it is a proposal that must be validated.

Authors' response: 

Regarding the title, we have deleted the phrase "and validated". In the discussion, we have added the following sentence: "We have also provided some provisional data on the validity of LiPADI, but due to a limited number of analyzed patients and underrepresentation of some subtypes of LP, these results have to be considered with caution and need to be verified in the future studies." In addition, in the original version of the manuscript, we have highlighted the study limitations, including a low number of patients, and a lack of some LP variants.